# Clinical Trials of Stem Cell Therapy in Japan: The Decade of Progress under the National Program

**DOI:** 10.3390/jcm11237030

**Published:** 2022-11-28

**Authors:** Shin Enosawa

**Affiliations:** Division for Advanced Medical Sciences, National Center for Child Health and Development, Tokyo 157-8535, Japan; enosawa-s@ncchd.go.jp

**Keywords:** stem cell, transplantation, clinical trial, iPS cell, ES cell, Japan, regulation, health insurance

## Abstract

Stem cell therapy is a current world-wide topic in medical science. Various therapies have been approved based on their effectiveness and put into practical use. In Japan, research and development-related stem cell therapy, generally referred to as regenerative medicine, has been led by the government. The national scheme started in 2002, and support for the transition to clinical trials has been accelerating since 2011. Of the initial 18 projects that were accepted in the budget for preclinical research, 15 projects have begun clinical trials so far. These include the transplantation of retinal, cardiac, and dopamine-producing cells differentiated from human induced pluripotent stem (iPS) cells and hepatocyte-like cells differentiated from human embryonic stem (ES) cells. The distinctive feature of the stem cell research in Japan is the use of iPS cells. A national framework was also been set-up to attain the final goal: health insurance coverage. Now, insurance covers cell transplantation therapies for the repair and recovery of damaged skin, articular cartilage, and stroke as well as therapies introduced from abroad, such as allogeneic mesenchymal stem cells for graft-versus-host disease and chimeric antigen receptor-T (CAR-T) cell therapy. To prepare this review, original information was sought from Japanese authentic websites, which are reliable but a little hard to access due to the fact of multiple less-organized databases and the language barrier. Then, each fact was corroborated by citing its English version or publication in international journals as much as possible. This review provides a summary of progress over the past decade under the national program and a state-of-the-art factual view of research activities, government policy, and regulation in Japan for the realization of stem cell therapy.

## 1. Introduction

Human whole genome sequencing and the progress in the elucidation of a biological reaction network have resulted in a new scientific term: bioinformatics. Currently, a number of biological events can be explained at the molecular level. The acceleration of research and development in life sciences has led to a global trend: the 21st century is the biotechnology era [1]. In Japan, the Biotechnology Strategy Council, established by the Cabinet Office of the Japanese government issued “Strategies for the Development of Biotechnology” in December 2002 as a vision of the 21st century [2,3,4]. This report proposed a new direction in science and technology, i.e., a shift from electronics to biotechnology. In response to this report, the Project for Realization of Regenerative Medicine was launched in 2003 [5].

Regenerative medicine originates from stem cell biology in embryology. The therapeutic concept is to restore the function of damaged organs and tissues by stem cells. The implantation or mobilization of stem cells shows unique potency not yet seen in existing pharmaceuticals, and the cell itself is being recognized as a new medicinal category. In particular, such therapy is expected to replace organ transplantation, which has a serious long-lasting donor shortage. In research and development, somatic stem cells and embryonic stem (ES) cells preceded and induced pluripotent stem cells (iPS) cells established by Shinya Yamanaka, a Nobel Prize awardee in 2012 [6,7]. iPS cells enable the use of autologous pluripotent stem cells, which are produced from somatic cells, and resolve not only the issue of the immunological rejection but also the ethical concerns of ES cells. The establishment of iPS cells was an epochal achievement of the biotechnology era in Japan.

In 2011, the Highway Program for Realization of Regenerative Medicine began, and it played a major role in the transition to clinical applications. The program is set under the Research Center Network for Realization of Regenerative Medicine, and the support for preclinical studies was enhanced. Of the 18 projects initially selected for the program, 15 projects have announced clinical implementation to date. In this review, the past and current status of clinical trials of stem cell therapy in Japan is discussed focusing on the progress under the national program.

## 2. Preclinical Research Projects Aiming at the Transition to Clinical Trials

The Highway Program for Realization of Regenerative Medicine and the Program of the Research Center Network for Realization of Regenerative Medicine recruited 18 scientific projects, 1 foundational platform, and 2 support programs (Table 1). The objective was to start clinical trials in 3 to 7 years. The scientific projects were categorized by the use of human somatic cells and the use of human iPS and ES cells. The former includes treatments for articular cartilage, corneal endothelium, liver cirrhosis, and intestinal epithelium, totaling five projects. The latter include treatments involving the eye (three projects), central nervous system (two projects), heart (two projects), liver (two projects), articular cartilage (one project), pancreatic islets (one project), and immune system (one project), totaling 13 projects, one of which involves the application of human ES cells and the rest the application of human iPS cells. Supporting programs cover regulation and ethical, legal, and social issues (ELSI), and the two teams organized regular meetings to instruct on key issues for the successful transition to clinical trials in a cross-sectoral manner.

Each project has its own originality based on the preceding research with somatic stem cells as well as human ES and iPS cells. Sekiya et al. first determined the superiority of mesenchymal stem cells isolated from synovium in chondrogenesis activity compared to stem cells from bone marrow, periosteum, skeletal muscle, and adipose tissue [8] and confirmed the repairing ability in a rabbit osteochondral defect model in vivo [9]. To obtain cells for transplantation of corneal endothelium, the disfunction of which induces a serious homeostatic imbalance of the anterior hydatoid, a ROCK inhibitor treatment was effective in triggering the cells’ proliferation, which was otherwise static [11]. The procedure enabled the novel treatment of cell engraftment for corneal endothelium disorder instead of corneal tissue transplantation [10]. The therapeutic potential of mesenchymal stem cells in the amelioration of cirrhotic liver has been investigated using autologous bone marrow cells [12]. However, it is often hard to harvest enough cells from patients with liver cirrhosis under general anesthesia. Thus, a method to increase the number of cells prior to transplantation was established [13]. In the mesenchymal stem cell therapy of knee joint cartilage, the preloading of magnetic beads with a medical grade helped to position the cells in the most effective place for the treatment [14]. The colonic stem cell-derived organoids showed the integration and repairment of an artificial ulcer in mouse colon, promising clinical application for inflammatory bowel diseases [15].

Most of the projects involving human iPS and ES cells are aimed at the transplantation of manufactured tissue to replace diseased tissue. All of the studies proposed the resolution of unmet medical needs by producing cells and tissues through quality controlled processing. For instance, the target disease of the retinal tissue regeneration was age-related macular degeneration [16,17]. There are a number of patients but still few effective treatments, except for the periodic topical injection of anti-vascular endothelial growth factor (VEGF) antibody to suppress retinal pathogenic thickening. Translocation of the perimacular area of a patient’s own retina to the diseased macular area was tried using intraocular microsurgery [36], but it is not conducted at present because of the invasiveness and lack of an obvious effect. Transplantation of stem cell-derived retinal tissue is hoped to be a permanent treatment, especially in order to escape the need for regular injections. In addition, there are also ethical rationales for conducting such clinical trials: (1) the disease is not life-threatening but highly impairs the quality of life; (2) the intraocular region is an immune privilege site; (3) carcinogenesis in the retina is very rare in adults. The project involving cell transplantation for Parkinson’s disease proposes to use human iPS cell-derived dopamine-producing cells [21]. A similar treatment was attempted with allogeneic fetal olfactory mucosal cells isolated from aborted fetus [37,38]. The procedure involved serious ethical issues and the effect was disputable. Olfactory cells contain not only dopamine- but also serotonin-secreting cells, which are not effective but rather aggravative toward neurological symptoms. The project aims to procure and transplant purified dopamine-producing cells under well-controlled cell processing manufacturing that is free of ethical concerns. Articular cartilage is commercially available [39] and its therapeutic potential is well known [40]. However, resource is limited because only cartilage from child and juvenile donors is effective. The human iPS cell-derived cartilage has characteristics of early developmental stages and is considered more suitable for cartilage repair [31]. In addition to the above-mentioned projects, clinical trial-oriented research on stem cell therapy for cornea [18,19], heart [20,28], liver [22,29], platelets [25], spinal cord [26], immune system (natural killer T cells) [30], and pancreatic islets [32] was started.

At the time of proposal, the projects using iPS cells were envisioned to use the iPS cells produced from the patient’s own somatic cells such as subcutaneous fibroblasts or lymphocytes. Later, the cell source was changed to the iPS cell stock as mentioned below (see Section 3.3) and all clinical trials are to use allogeneic iPS cells except for Masayo Takahashi’s first trial and Koji Eto’s platelet trial (6_-1_ and 11 in Table 2).

## 3. Social Framework for Clinical Trials of Stem Cell Therapy

### 3.1. Legislation

Along with the increased support in the research budgets, the adjustment of regulations is important for conducting a national scheme. Previously, only an explanatory clinical run of a cell therapy was allowed, under a decision by the head of the institution in accordance with a report by the institutional review board, which was installed based on national ethics guidelines. At that time, the data were not valid for official investigations of new drugs. In 2014, the revised Act on Securing Quality, Efficacy and Safety of Products Including Pharmaceuticals and Medical Devices (Act No.145 of 10 August 1960) was enacted [41] and it included two new categories: regenerative medical products and gene therapy products. Regenerative medical products consist of human somatic stem cell- and human somatic cell-processed products at present, because there are no approved products with cells derived from iPS and ES cells. Characteristically, a system of conditional and time-limited approval was introduced for regenerative medical products. The purpose of this system is to apply the products to patients if they are considered to be effective, since it often takes time to determine the effectiveness of regenerative medicine. Once approval or conditional and time-limited approval is given, the medical treatments with the products are covered by health insurance.

In 2014, the Act on the Safety of Regenerative Medicine (Act No.85 of 27 November 2013) was enforced [42]. According to this law, the Ministry of Health, Labor, and Welfare examines all protocols of clinical research on regenerative medicine that had been regulated under national ethics guidelines before. The pancreatic islet transplantation was first performed under this law and after the data estimation was completed, the procedure was authorized and covered by health insurance in 2021, as introduced by Noguchi [43].

### 3.2. Japan Agency for Medical Research and Development (AMED)

The Japan Agency for Medical Research and Development (AMED) was established in 2015, as the headquarter that play a central role in the promotion of research and development in the medical field and transition to commercialization [44]. The budget for clinical-oriented research and development, which had been allocated independently by the Ministry of Health, Labor and Welfare, the Ministry of Education, Culture, Sports, Science and Technology, and the Ministry of Economy, Trade, and Industry, is now allocated by the AMED. The AMED also maneuvers the approved research projects for their satisfactory execution.

### 3.3. iPS Cell Stock Project

The advantage of iPS cells in stem cell therapy is the availability of autologous iPS cells, which escape from allogeneic immune rejection. However, the establishment of an iPS cell line is extremely time and cost consuming, especially to establish clinical-grade iPS cell lines that are secured by good manufacturing practice (GMP). Moreover, the acquirement of a successful iPS cell line is affected by individual donor differences, including age. Therefore, the Program of the Research Center Network for Realization of Regenerative Medicine shifted to the use of allogeneic iPS cell lines from tailor-made autologous iPS cells. The iPS cell stock project embarked to produce a collection of iPS cell lines from persons whose human leukocyte antigen (HLA) haplotype is homozygous [33,34,35]. As far as HLA-A, HLA-B, and HLA-DR are concerned, the compatibility of those homozygous cells is wider than for heterozygous cells, and 140 homohaplotype lines will cover 90% of the Japanese population [35]. Although those cells are still allogeneic in minor histocompatibility antigens, the rejection reaction is thought to be mild. Currently, GMP-secured homohaplotype iPS cell lines are used in preclinical studies that are in the middle to late stages when clinical trials are within range.

## 4. Clinical Trials That Transitioned from Preclinical Projects

Table 2 displays a list of the clinical trials that were derived from the projects shown in Table 1. All projects with somatic stem cells have started, and some results have been published. The project involving the repair of articular cartilage has four subsidiary trials (1_-1_, 1_-2_, 1_-3_, and 1_-4_) from slightly different perspectives. Likewise, the projects involving corneal and liver regeneration have set four and three protocols, respectively. The clinical trials listed here were started after the abovementioned two laws were enacted in 2014, and some similar clinical studies were conducted before then under national ethics guidelines.

As for iPS cells and ES cells, 10 out of 13 of the projects have started clinical trials. Although projects with iPS and ES cells are a little behind compared to somatic stem cells, the first-in-human tests are progressing smoothly. The most advanced is the retinal regeneration trial, and the results of two cases with autologous iPS cells (6_-1_) and results with five cases using allogeneic iPS cells (6_-2_) have already been published [45,46]. The original project was designed with autologous iPS cells, but due to the start of the iPS stock project, the clinical trial shifted toward the use of allogeneic iPS cells. Therefore, the trial with the autologous iPS cells was finished after the completion of the two cases, while the target sample size was five.

**Table 2 jcm-11-07030-t002:** List of clinical trials that were derived from the preclinical projects in Table 1.

No. in Table 1 ^1^	Disclosure Date	Project Title, Source of Cells ^2^, (Registration ID ^3^), Last Follow-Up Date	Phase	Sample Size	Report of First-In-Human Test
Projects with human somatic stem cells					
1_-1_	1 October 2013	Healing acceleration of repaired meniscus by synovial stem cells “Clinical study to assess the safety and efficacy of transplantation of autologous synovial mesenchymal stem cells in patients with knee meniscal tear”, Autologous articular synovial tissue, (UMIN000011881), 21 April 2015	Safety, Efficacy	5	Published [45]
1_-2_	12 June 2015	Enhancement of healing for extruded injured meniscus by transplantation of synovial stem cells after meniscus surgery, Autologous articular synovial tissue, (UMIN000017890), 31 May 2016		10	Published [45]
1_-3_	1 May 2017	Investigator initiated clinical trial of autologous synovial stem cells for meniscus lesions, Autologous articular synovial tissue, (UMIN000026383), 21 February 2019	I	10	
1_-4_	1 June 2017	Intraarticular injections of synovial stem cells for osteoarthritis of the knee, Autologous articular synovial tissue, (UMIN000026732), 10 March 2020	I	10	Published [46,47]
2_-1_	9 December 2013	Clinical application of corneal endothelial regenerative medicine by means of cultured human corneal endothelial cell transplantation, Allogeneic corneal endothelium, (UMIN000012534), 31 May 2023		45	Published [48]
2_-2_	21 July 2017	Investigator initiated trial of cultivated human corneal endothelial cell injection, Allogeneic corneal endothelium, (UMIN000028324), 19 November 2018	II	15	
2_-3_	1 October 2018	Investigator initiated confirmatory trial of cultivated human corneal endothelial cell injection, Allogeneic corneal endothelium, (UMIN000034334), 8 July 2019	III	12	
2_-4_	5 April 2019	Clinical application of corneal endothelial regenerative medicine by means of cultured human corneal endothelial cell transplantation: long-term follow up, Allogeneic corneal endothelium, (UMIN000036422), 31 March 2024		45	Published [48,49]
3_-1_	2 March 2015	Safety study of a less invasive liver regeneration therapy using cultured autologous bone marrow-derived mesenchymal stem cells for decompensated liver cirrhotic patient, Autologous bone marrow, (UMIN000016686), date not listed	I	10	Conference presentation 16 April 2016
3_-2_	15 January 2019	Study on the safety of hepatic arterial infusion of cultured autologous bone marrow cells in patients with decompensated liver cirrhosis, Autologous bone marrow, (UMIN000035528), 19 August 2021	I	5	
3_-3_	31 August 2020	An open-label, uncontrolled study to evaluate the efficacy and safety of autologous bone marrow mesenchymal stem cells (LS-ABMSC1) in patients with decompensated liver cirrhosis, Autologous bone marrow, (Phase I/II study) (UMIN000041461), 17 February 2023	I/II	10	
4	5 October 2021	Investigator-initiated clinical trial to evaluate the safety and efficacy of AX-1911, magnetic field generator for magnetic targeting, in knee osteoarthritis, Autologous bone marrow, (jRCT2062210039), 30 September 2022	I	5	
5	1 April 2018	Mucosal regeneration therapy by autologous intestinal stem cell transplantation to inflammatory bowel disease patients, Autologous intestinal tissue, (UMIN000030117), date not listed	Safety	8	Press release 6 July 2022
Projects with human iPS or ES cells					
6_-1_	2 October 2013	A study of transplantation of autologous induce d pluripotent stem cell (iPSC) derived retinal pigment epithelium (RPE) cell sheet in subjects with exudative age-related macular degeneration, Autologous skin fibroblasts, (UMIN000011929), 28 February 2019	Exploratory	2 (5) ^4^	Published [50]
6_-2_	6 February 2017	A study of transplantation of allogenic induced pluripotent stem cell (iPSC) derived retinal pigment epithelium (RPE) cell suspension in subjects with neovascular age related macular degeneration, HLA homozygous allogeneic iPS cells from the iPS cell stock, (UMIN000026003), 20 September 2021	Exploratory	5	Published [51]
7	23 May 2019	First-in-human clinical research of iPS derived corneal epithelial cell sheet transplantation for patients with limbal stem-cell deficiency, HLA homozygous allogeneic iPS cells from the iPS cell stock, (UMIN000036539), 13 December 2021	I	4	Press release 29 August 2019
8	1 April 2022	A prospective observational study of induced pluripotent stem cell-derived cardiac spheres transplantation EXTENDed Follow-up, HLA homozygous allogeneic iPS cells from the iPS cell stock, (UMIN000047335), 31 March 2029	Safety, Efficacy	3	
9	12 September 2018	Kyoto trial to evaluate the safety and efficacy of iPSC-derived dopaminergic progenitors in the treatment of Parkinson’s disease, HLA homozygous allogeneic iPS cells from the iPS cell stock, (UMIN000033564), 31 December 2023	I/II	7	Press release 9 November 2018
10	27February 2019	Clinical study of HAES transplantation in patients with neonatal onset urea cycle disorder, Allogeneic human ES cells, SEES series [23,24], (JMA-IIA00412), 30 September 2022	I/II	5	Press release 21 May 2020
11	5 March 2020	Clinical study of autologous transfusion of iPS cell-derived platelets for thrombocytopenia (iPLAT1), Autologous peripheral blood mononuclear cells, (jRCTa050190117), 20 January 2021		1	Published [52]
12	1 December 2020	Regenerative medicine for spinal cord injury at subacute stage using human induced pluripotent stem cell-derived neural stem/progenitor cells, HLA homozygous allogeneic iPS cells from the iPS cell stock, (UMIN000035074), 30 November 2023	Safety	4	
14_-1_	11 December 2019	Clinical trial of human (allogeneic) iPS cell-derived cardiomyocytes sheet for ischemic cardiomyopathy, HLA homozygous allogeneic iPS cells from the iPS cell stock, (jRCT2053190081), 30 May 2024	I	10	Press release 25 December 2020
14_-2_	22 June 2022	Clinical trial of human (allogeneic) iPS cell-derived cardiomyocytes sheet for ischemic cardiomyopathy (Follow-up trial), HLA homozygous allogeneic iPS cells from the iPS cell stock, (jRCT2053220055), 31 March 2029	I	10	
16	11 September 2020	A Phase I study of iPS-NKT cell intra-arterial infusion therapy in patients with recurrent or advanced head and neck cancer (First in human study), Allogeneic iPS cells from NKT cells of healthy volunteers, (jRCT2033200116), 31 March 2022	I	9	Press release 29 June 2020
17	7 February 2020	A clinical study for treatment of articular cartilage damage in knee joints with allogeneic induced pluripotent stem (iPS) cell-derived cartilage (TACK-iPS), HLA homozygous allogeneic iPS cells from the iPS cell stock, (jRCTa050190104), 31 December 2023		4	

^1^ The numbers correspond to those in Table 1. The numbers in subscript indicate another clinical study derived from the same preclinical project. ^2^ Cell sources confirmed in the literature or websites of clinical trial registries were noted. ^3^ Each protocol is registered in UMIN (https://www.umin.ac.jp/english/, accessed on 17 October 2022), jRCT (https://jrct.niph.go.jp/search, accessed on 17 October 2022), or JMA (http://www.jmacct.med.or.jp/en/what-we-do/registry.html, accessed on 17 October 2022). ^4^ In the original protocol, the sample size was set at five, but because of the policy shift toward the use of allogeneic iPS cells, the project was finished when two cases were completed. For more details, see Appendix A where the URL of each project is listed.

Another noteworthy clinical trial is the autologous cardiocyte transplantation for neonates with fatal congenital heart disease by Hidemasa Oh at Okayama University Hospital [53]. His strategy is to return patient’s own cells isolated from tissue that was removed during neonatal cardioplasty to the cardiac tissue. The clinical trials are registered at ClinicalTrials.gov. as the TICAP prospective phase 1, PERSEUS randomized phase 2, APOLLON phase 3 randomized multicenter clinical trial, and the TICAP-DCM study.

## 5. Approved Stem Cell Therapies

Following the enforcement of the revised Act on Securing Quality, Efficacy and Safety of Products Including Pharmaceuticals and Medical Devices, 13 cell-processed products have been approved (Table 3). All products are based on the use of somatic cells or somatic stem cells, not on the use of iPS and ES cells yet.

Japan Tissue Engineering is a pioneer company in the development of cell therapy in Japan and has two approved products, JACE and JACCthat have been developed as an autologous artificial skin for severe heat burn [54] and cartilage defects [55], respectively. Those two products were first approved in 2007 and 2012, respectively as a medical device in 2012, when the former unamended act had no category for the approval of regenerative medical products. After the reexamination for the expansion of indications, both were designated as human somatic stem cell-processed products.

Chimeric antigen receptor-T (CAR-T) cell products and TEMCELL (cellular drug for acute graft-versus-host disease after allogeneic hematopoietic stem cell transplantation) were developed abroad and introduced.

After the revision of the act, TEMCELL and HeartSheet were the first items approved as regenerative medical products in 2015. HeartSheet is a cell sheet of autologous skeletal myoblast cultured on thermoresponsive hydrogels [57]. The sheets are applied on the heart surface of the patient with ischemic heart failure to induce revascularization in necrotic myocardial tissue. Nature dealt with the conditional approval of HeartSheet in an editorial [58]. Approval was given with seven cases, and it is expected that the number of cases will be increased to 60. The editorial posed whether it is acceptable to impose medical expenses to patients at this stage. It is hoped that the validity will be determined soon together with iPS cell-derived cardiomyocyte sheets.

## 6. Next Generation Research in Stem Cell Therapy

Preclinical research is progressing steadily. The transfer of genes such as LOTUS or DREADDs have enhanced the ability to repair spinal cord injuries using human iPS cell-derived neural stem cells [59,60]. Along with the central nervous system, effective stem cell therapy for peripheral nerve regeneration has been reported using a bio three-dimension printer and xeno-free medium [61]. Approaches using stem cell therapy for the kidney and lung are also advancing. Transplantation of pig kidneys into monkeys requires strong immunosuppression but fetal porcine kidney survived with milder clinically applicable immunosuppression [62]. The transplanted kidney was vascularized by the host blood vessels, and this may work as an embryonic organ complementation site. A kidney precursor tissue, ureteric bud, was generated from human iPS cells [63]. The human iPS cell-derived lung progenitors were successfully integrated into mice lungs [64]. In addition, human tracheal tissue was created in vitro by combining cartilage, mesenchyme, and smooth muscle differentiation [65,66]. A new aspect of stem cell therapy is its use in the treatment of brain tumors. Glioblastoma is characterized by diffuse infiltration into the normal brain. Human iPS cells expressing a suicide gene had higher tumor-trophic migratory capacity and antitumor effects, indicating the potentiality of iPS cell-based therapy for invasive glioblastoma [67]. Induced tissue-specific stem (iTS) cells and dental pulp cells are unique and promising cells for therapy. Similar to iPS cells, iTS cells are produced by transient overexpression of the reprogramming factors but they still possess differentiation directivity to original cells, due to epigenetic memory [68]. iTS cells do not show full pluripotency, but have excellent differentiation directionality. Neural crest-origin dental pulp cells are attracting attention as a stem cell source [69]. Those cells obtained from deciduous teeth and young wisdom teeth are considered to have fewer ethical problems and higher stemness. These studies will be applied for clinical trials in the next decade.

## 7. Conclusions

A national program to support the transition of stem cell therapy into clinical practice has been in operation since 2011 in Japan, and 15 out of the 18 initial projects have started clinical trials. Along with the research budget, social frameworks, such as legislation, a research headquarters, and an iPS cell stock were established. There are currently 13 approved cell-processed products. Cell therapy will face judgement on not only its effectiveness but also economic efficiency. Over the past decade, the social infrastructure for the development to the commercialization of regenerative medical products was established together with global harmonization. It is hoped that better products will be created in the near future based on this platform.

## Figures and Tables

**Table 1 jcm-11-07030-t001:** List of initial preclinical research projects that were adopted in the first 3 years of the Program of the Research Center Network for Realization of Regenerative Medicine (projects started in 2011 belonged to the Highway Program for Realization of Regenerative Medicine). Compiled by the author based on [5].

No.	Inaugural Year	Project Title, Source of Cells ^1^, Representative Publication Related to the Proposal	Principal Investigator	Affiliation ^2^
Projects with human somatic stem cells				
1	2011	Meniscal regeneration in the knee using synovial stem cells, Autologous articular synovial tissue, [8,9]	Ichiro Sekiya	Tokyo Medical and Dental University
2	2011	Clinical application of corneal endothelial regenerative medicine by means of cultured human corneal endothelial cell transplantation, Allogeneic corneal endothelium, [10,11]	Shigeru Kinoshita	Kyoto Prefectural University of Medicine
3	2011	Development of a less invasive liver regeneration therapy using cultured human bone marrow derived cells, Autologous bone marrow, [12,13]	Isao Sakaida	Yamaguchi University
4	2012	Bone and cartilage regeneration using magnetic targeting system of magnetically labeled bone marrow mesenchymal cells, Autologous bone marrow, [14]	Mitsuo Ochi	Hiroshima University
5	2013	Center for development of mucosal regenerative therapies for inflammatory bowel diseases using cultured intestinal epithelial stem cells, Autologous colonic crypts, [15]	Mamoru Watanabe	Tokyo Medical and Dental University
Projects with human iPS ^3^ and ES cells				
6	2011	Development of methods for treating age-related macular degeneration by transplantation of retinal pigment epithelial (RPE) cells derived from induced pluripotent stem (iPS) cells [16,17]	Masayo Takahashi	RIKEN ^4^
7	2011	Development of corneal regenerative treatment methods using iPS cells [18,19]	Koji Nishida	Osaka University
8	2011	Establishment of regenerative therapies for severe heart failure by transplantation of iPS cells-derived cardiomyocytes [20]	Keiichi Fukuda	Keio University
9	2011	Development of cell replacement therapy using iPS cell-derived neural cells against Parkinson’s disease and stroke [21]	Jun Takahashi	CiRA ^5^
10	2011	Clinical research on human embryonic stem (ES) cell formulations for treatment of congenital metabolic disorders giving rise to severe hyperammonemia, Allogeneic ES cells (SEES series) [22,23,24]	Akihiro Umezawa	National Center for Child Health and Development
11	2012	Development of and clinical studies on platelet preparations based on induced pluripotent stem (iPS) cell techniques [25]	Koji Eto	CiRA
12	2013	Regenerative medicine for spinal cord injury and stroke using neural precursor cells of iPS cell origin [26]	Hideyuki Okano	Keio University
13	2013	Research and development center for clinical application of complex tissue formation technologies to restore visual function [27]	Masayo Takahashi	CiRA
14	2013	Center for the development of myocardial regenerative treatments using iPS cells [28]	Yoshiki Sawa	Osaka University
15	2013	Center for development of innovative technologies for metabolic organs using induced pluripotent stem (iPS) cells [29]	Hideki Taniguchi	Yokohama City University
16	2013	Center for development of cancer immunotherapy technology by regenerating natural killer T-cells (NKT cells) [30]	Haruhiko Koseki	RIKEN
17	2013	Center for development of regenerative therapies for cartilage diseases using induced pluripotent stem (iPS) -cell-derived chondrocytes [31]	Noriyuki Tsumaki	CiRA
18	2013	Center for development of next-generation pancreatic islet transplantation methods based on induced pluripotent stem (iPS) cell technology [32]	Atsushi Miyajima	University of Tokyo
Core Center for iPS Cell Research				
19	2013	Center of excellence in development of iPS cell stock for regenerative medicine, Mononuclear cells of peripheral or umbilical cord blood from donors of which human leukocyte antigen (HLA) haplotype is homozygous [33,34,35]	Shinya Yamanaka	CiRA
Regulation support				
20	2011	Support for research and development with the aim of early-stage realization and overseas expansion of regenerative medicine	Akifumi Matsuyama	National Institute of Biomedical Innovation
Ethical support				
21	2011	Research on the ethical, legal and social implications related to regenerative medicine	Kaori Muto	University of Tokyo

^1^ Source of cells at the time of proposal; ^2^ Affiliation at time of adoption; ^3^ The projects using iPS cells were envisioned to use the iPS cells produced from the patient’s own somatic cells such as subcutaneous fibroblasts or lymphocytes. ^4^ RIKEN: Institute of Physical and Chemical Research (https://www.riken.jp/en/, accessed on 17 October 2022); ^5^ CiRA: Center for iPS Cell Research and Application, Kyoto University (https://www.cira.kyoto-u.ac.jp/e/index.html, accessed on 17 October 2022).

**Table 3 jcm-11-07030-t003:** List of regenerative medical products that have been approved by the Japanese government under the Act on Securing Quality, Efficacy and Safety of Products Including Pharmaceuticals and Medical Devices (compiled by the author based on the website of the Pharmaceuticals and Medical Devices Agency [56]).

No.	Approval Date	Nonproprietary Name, Material, and Indication	Brand Name (Company)	Decision
Human somatic cell-processed products				
1	29 September 2016, 28 December 2018	Human autologous epidermis-derived cell sheet Material: Patient’s own healthy skin tissue co-cultured with mouse embryo-derived 3T3-J2 feeder cells Indication: Severe heat burn, giant congenital melanocytic nevus, dystrophic epidermolysis bullosa, and junctional epidermolysis bullosa	JACE (Japan Tissue Engineering Co., Ltd. Gamagori, Japan)	Change approved
2	26 March 2019	Tisagenlecleucel Material: T cells derived from patient’s peripheral blood Indication: CD19-positive relapsed or refractory B-cell acute lymphoblastic leukemia and CD19-positive relapsed or refractory diffuse large B-cell lymphoma	Kymriah Suspension for Intravenous Infusion (Novartis Pharma K.K. Tokyo, Japan)	Approval
3	22 January 2021	Axicabtagene ciloleucel Material: T cells derived from patient’s peripheral blood Indication: Relapsed or refractory large B-cell lymphoma	YESCARTA Intravenous Drip Infusion (Daiichi Sankyo Company, Limited, Tokyo, Japan)	Approval
4	22 March 2021	Lisocabtagene maraleucel Material: T cells derived from patient’s peripheral blood Indication: Relapsed or refractory large B-cell lymphoma and relapsed or refractory follicular lymphoma	Breyanzi Suspension for Intravenous Infusion (Celgene Corporation, Tokyo, Japan)	Approval
5	20 January 2022	Idecabtagene vicleucel Material: T cells derived from patient’s peripheral blood Indication: Relapsed or refractory multiple myeloma	Abecma Intravenous Infusion (Bristol-Myers Squibb K.K. Tokyo, Japan)	Approval
6	20 June 2022	Human autologous tissue for transplantation Material: Patient’s own cartilage tissue mixed with gel-form atelocollagen Indication: Traumatic cartilage defects or osteochondritis dissecans of the knee excluding knee osteoarthritis	JACC (Japan Tissue En-gineering Co., Ltd. Gamagori, Japan)	Change approved
Human somatic stem cell-processed products				
7	18 September 2015	Human allogeneic bone-marrow-derived mesenchymal stem cell Material: Human allogeneic bone-marrow-derived mesenchymal stem cell from healthy adult donor Indication: Acute graft-versus-host disease after allogeneic hematopoietic stem cell transplantation	TEMCELL HS Inj. (JCR Pharmaceuticals Co., Ltd. Ashiya, Japan)	Approval
8	18 September 2015	Human autologous skeletal myoblast-derived cell sheet Material: Patient’s own skeletal myoblasts that were cultured in sheet form using temperature-responsive polymer, poly(N-isopropylacrylamide) (PIPAAm) Indication: Serious heart failure caused by ischemic heart disease when standard therapies are not sufficiently effective	HeartSheet (Terumo Corporation, Tokyo, Japan)	Conditional time-limited approval
9	28 December 2018	Human autologous bone-marrow-derived mesenchymal stem cell Material: Patient’s own bone-marrow-derived mesenchymal stem cells Indication: Spinal cord injury only for use in patients with traumatic spinal cord injury and ASIA Impairment Scale A, B, or C	STEMIRAC Inj. (Nipro Corporation, Osaka, Japan)	Conditional time-limited approval
10	19 March 2020	Human autologous corneal limbus-derived corneal epithelial cell sheet Material: Patient’s own corneal epithelial cells from limbal tissue that were cultured in sheet form using temperature-responsive polymer, PIPAAm Indication: Limbal stem cell deficiency	Nepic (Japan Tissue Engineering Co., Ltd. Gamagori, Japan)	Approval
11	11 June 2021	Human autologous oral mucosa-derived epithelial cell sheet Material: Patient’s own oral mucosal epithelial cells that were cultured in sheet form using temperature-responsive polymer, PIPAAm Indication: Repair of corneal epithelium defects onto the ocular surface of patients with limbal stem cell deficiency	Ocural (Japan Tissue Engineering Co., Ltd. Gamagori, Japan)	Approval
12	27 September 2021	Darvadstrocel Material: Human allogenic adipose tissue-derived stem cells from subcutaneous adipose tissue of healthy adult donor Indication: Crohn’s disease	Alofisel Injection (Takeda Pharmaceutical Company Limited, Osaka, Japan)	Approval
13	20 January 2022	Human autologous oral mucosa-derived epithelial cell sheet using a human amniotic membrane substrate Material: Patient’s own oral mucosal epithelial cells that are cultured on allogeneic amniotic membrane substrate Indication: Limbal stem cell deficiency	Sakracy (Hirosaki Lifescience Innovation, Inc. Hirosaki, Japan)	Approval

## Data Availability

Not applicable.

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
