# Peer review of "Clinical Trials of Stem Cell Therapy in Japan: The Decade of Progress under the National Program"

_jcm, 2022, doi:10.3390/jcm11237030_

Round 1
Reviewer 1 Report
1. The preclinical studies only from 2011 to 2013? I think it may be better to add the research progress in recent years, which may involve more types of stem cells, stem cell derivatives and therapeutic range. Limited to the Highway Program for Realization of Regenerative Medicine in the first 3 years, which was launched in 2011, I think its innovation and guidance are not outstanding enough. The first three years of preclinical and clinical studies can be emphasized, but it is recommended to introduce the progress of preclinical and clinical studies in recent years, and point out the future development trends and prospects.
2. The paper describes related preclinical studies, clinical trials and regenerative medical products that have been approved since the highway program was launched in Japan. It is a complete process, but more like a data statistics, I think it is better to do some analyses and summaries . Such as preclinical research mainly adopts what animal model, what disease treatment is proved to be effective or uneffective, the clinical research on stem cells mainly use what means to intervene.
The author mainly described the human cells and iPSC, in my opinion, it is not comprehensive enough, and the analyses and summaries about studies on each part are not enough, And it lacks the prospect of the future development of stem cells in Japan and what laws and regulations should be paid attention to. that is, the depth is not enough to support the topic.
I don't think it can be seen from the paper that clinical trials of stem cell therapy in Japan has made great progress, nor can it get enough inspiration.
I think it is better to improve.
3. I suggested that each product should specify what cell is used in the list of regenerative medical products.
Reviewer 2 Report
Comments:
1) Introduction should be improved. Current text in this section does not provide sufficient introductory information for readers. The author has to write more information on stem cell therapy. In the end of introductory part, I recommend to add for example, “In this review, we discuss current status of clinical trials of stem cell therapy in Japan…
2) In Table 2, I recommend you to insert the following additional information: Number of Patients, Route of Administration and Doses, Follow-Up Time, Clinical Status before and after treatment. I think this information will be very useful for readers (clinicians and researchers) who are interested in stem cell therapy and clinical trials.
Reviewer 3 Report
The Review submitted by S. Enosawa is well written and provides a useful overview for future research. Minor issues should be addressed prior acceptance:
p7, line 146-147: is there any information when the cited ongoing clinical trials will be finished? A time-frame (approx.) might be helpful.
p7, line 154: please include a reference to Table 3 to explain the terms JACE and JACC
p9 conclusions: I recommend to include a brief outlook.
Round 2
Reviewer 1 Report
Much better. I think there are the following points that can be improved:
1、In this review, past and current status of clinical trials of stem cell therapy in Japan is discussed focusing on the progress in a national program.I think it is better to state in the title or abstract that the stem cell research project listed in this paper is based on the Highway Program for Realization of Regenerative Medicine launched in 2011. For example, the title of the article should be changed to: "Clinical Trials of Stem Cell Therapy in Japan:Under the national program", Instead of all the clinical Trials in Japan in the past ten years, so as not to let readers misunderstand.
2、Table 1 and Table 2 also explain the source of stem cells used, which body cell is the source of iPS cell, so that readers can better understand and compare.
Author Response
Dear Reviewer 1
Thank you very much for your considerate comments. I revised the manuscript according to your instruction.
Point-by-point responses to Reviewer 1
1. In this review, past and current status of clinical trials of stem cell therapy in Japan is discussed focusing on the progress in a national program. I think it is better to state in the title or abstract that the stem cell research project listed in this paper is based on the Highway Program for Realization of Regenerative Medicine launched in 2011. For example, the title of the article should be changed to: "Clinical Trials of Stem Cell Therapy in Japan: Under the national program", Instead of all the clinical Trials in Japan in the past ten years, so as not to let readers misunderstand.
Thank you very much for your helpful suggestion. According to your kind exemplification, I added “under the national program” in the title. I would be grateful if you allow me to retain “the decade of progress” that will clarify the term of which the manuscript covers. In addition, I added “under the national program” in the abstract and the introduction (serial line number 24 and 58, respectively).
2. Table 1 and Table 2 also explain the source of stem cells used, which body cell is the source of iPS cell, so that readers can better understand and compare.
Thank you very much for your thoughtful comment. It will be a great help for readers. I added the cell source in each project in Table 1 and Table 2.
As for Table 1, the projects using iPS cells was envisioned to use the iPS cells produced from the patient's own somatic cells such as subcutaneous fibroblasts or lymphocytes at the time of proposal. I added the explanation in the note and text (line 76-77 and line 127-131). In addition, to specify the property of human ES cells, I added references 23 and 22.
As for Table 2, I rechecked literatures and clinical trial registries carefully and added the source of the cells. In the clinical trials with human iPS cells, two projects used autologous iPS cells derived from somatic cells; skin fibroblasts in clinical trial 6-1 and peripheral blood in clinical trial 11, and others used allogeneic iPS cells.
Other revisions by self-check
1. I checked throughout the manuscript carefully and corrected misspellings and typos. (line 41, 64, 178, 245)
2. I am very sorry that I noticed that the order was wrong in Table 2 and supplementary table. I fixed them in the correct order.
